# Colonization with extended-spectrum β-lactamase and carbapenemase-producing *Enterobacterales* in Ethiopia: A systematic review and meta-analysis

**Mitkie Tigabie**[ID][1]*, **Getu Girmay**[2], **Yalewayker Gashaw**[3], **Getachew Bitew**[ID][1], **Abebe Birhanu**[ID][1], **Eden Getaneh**[1], **Azanaw Amare**[1], **Muluneh Assefa**[1]

**1** Department of Medical Microbiology, School of Biomedical and Laboratory Sciences, College of Medicine and Health Sciences, University of Gondar, Gondar, Ethiopia, **2** Department of Immunology and Molecular Biology, School of Biomedical and Laboratory Sciences, College of Medicine and Health Sciences, University of Gondar, Gondar, Ethiopia, **3** Department of Medical Laboratory Sciences, College of Health Sciences, Woldia University, Woldia, Ethiopia

* mitku1621@gmail.com

## Abstract

### Background

The human intestinal tract contains many commensals. However, during an imbalance of the normal microbiota following exposure to antibiotics, extended-spectrum β-lactamase- and carbapenemase-producing *Enterobacterales* emerge. Individuals colonized with these bacteria may develop subsequent infections themselves. Therefore, this review aimed to estimate the colonization rate of extended-spectrum β-lactamase- and carbapenemase-producing *Enterobacterales* in Ethiopia.

### Methods

The protocol was registered (PROSPERO ID: CRD42024550137). A systematic literature search was conducted in electronic databases, including PubMed, Google Scholar, and Hinari, to retrieve potential studies. The quality of the included studies was assessed using the Joanna Briggs Institute critical appraisal tool. The data were extracted from the eligible studies using Microsoft Excel 2019 and analyzed using STATA version 11. Heterogeneity between studies was checked using $I^2$ test statistics. Publication bias was assessed using funnel plots and Egger's test. A random-effects model of DerSimonian-Laird method was employed to estimate the outcomes.

### Results

A total of 15 studies with 4713 participants were included in the meta-analysis. The overall pooled colonization rates of extended-spectrum β-lactamase-producing and carbapenemase-producing *Enterobacterales* in Ethiopia were 28.5% (95% CI: 16.4-40.5%, $I^2$ = 95.9%, p < 0.001) and 4.4% (95% CI: 0.9–7.9%, $I^2$ = 0.0%, p = 0.64), respectively. The majority of the extended-spectrum β-lactamase producers were *E. coli* (20.6%,

**Data availability statement:** All relevant data are within the paper and its Supporting Information files.

**Funding:** The author(s) received no specific funding for this work.

**Competing interests:** All authors declare no conflicts of interest.

**Abbreviations:** CDT, combination disk test; CPE, carbapenemase-producing *Enterobacterales*; DDST, double-disk synergy test; ESBL, extended-spectrum β-lactamase; ESBL-PE, extended-spectrum β-lactamase-producing *Enterobacterales*; MDR, Multidrug resistance.

95% CI: 9.3–31.9%, I² = 94.4%, p < 0.001), followed by *Klebsiella* spp. (11.1%, 95% CI: 7.7–14.6%, I² = 20.2%, p = 0.245). Similarly, the predominant carbapenemase producers were *E. coli* (2.7%, 95% CI: -1.3–6.7, I² = 0.0%, p = 0.941) and *Klebsiella* spp. (2.1%, 95% CI: -1.7–5.9%, I² = 0.0%, p = 0.999). Furthermore, the pooled estimate of multidrug resistance among extended-spectrum β-lactamase producers was 71.7% (95% CI: 55.25–88.05%, I² = 92.9%, p < 0.001).

## Conclusion and recommendations

Approximately one-quarter of Ethiopians are colonized with ESBL-PE, while about one in 25 is colonized with CPE. These findings were obtained from studies with a moderate-to-low risk of bias. However, the results for ESBL-PE showed significant variability, indicating high heterogeneity among the studies. This colonization may lead to subsequent extraintestinal infections. Therefore, proactive action from all stakeholders is required to combat the unrecognized spread of extended-spectrum β-lactamase- and carbapenemase-producing *Enterobacterales* in humans.

## Introduction

*Enterobacterales* are a heterogeneous group of Gram-negative rods. The principal habitat of many of these bacteria is the gastrointestinal tract of humans and animals [1]. Currently, Enterobacterales are resistant to multiple antibiotics, including last-resort antibiotics, due to the production of beta-lactamase enzymes. These enzymes are major causes of bacterial resistance to the beta-lactam family of antibiotics, such as penicillins, cephalosporins, and carbapenems. As these enzymes are commonly produced by members of *Enterobacterales,* these bacteria are considered significant threats to human health [2].

In particular, extended-spectrum β-lactamase-producing *Enterobacterales* (ESBL-PE) and carbapenemase-producing *Enterobacterales* (CPE) are major global threats. In May 2024, the World Health Organization released ESBL-PE and CPE within the critical priority category of pathogens due to their high burden, ability to resist treatment and ability to spread resistance to other bacteria [3].

Gastrointestinal colonization with ESBL-PE and CPE is asymptomatic and can result in unrecognized spread to noncolonized individuals [4]. Moreover, after chronic carriage of ESBL-PE and CPE, patients develop severe, hard-to-treat infections, such as infections of the wound, respiratory tract, or urinary tract, and bloodstream infections during antimicrobial treatment, immunosuppression, imbalance of the normal microbiota, trauma, and invasive procedures such as surgery [4,5]. This is supported by S. Hess et al., as prior colonization with antibiotic-resistant Gram-negative bacteria was strongly associated with subsequent antibiotic-resistant bacteremia in immune-compromised patients [6].

The extended-spectrum β-lactamase and carbapenemase genes are plasmids, and transposon-mediated genes can exhibit co-resistance to many other classes of antibiotics and spread easily to other species of bacteria [7]. Both hospitalized patients and healthy individuals may be colonized by ESBL-PE and CPE and develop subsequent infections. Healthy individuals are important reservoirs for ESBL-PE [8]. However, previous hospitalization, a history of invasive procedures and the use of antibiotics for three or more months are risk factors for ESBL-PE colonization [9], and a history of international travel is also correlated with ESBL-PE colonization [8].

In developing countries such as Ethiopia, weak health systems, environmental contamination, poor water, poor sanitation, poor hygiene infrastructure and poor practices can increase the prevalence of ESBL-PE and CPE [10]. Additionally, in Ethiopia, the overuse and misuse of antimicrobial agents in the environmental sector, agriculture, and veterinary medicine drive the emergence and spread of ESBL-PE and CPE [11,12].

Meta-analysis examining colonization rates of ESBL-PE and CPE in fecal specimens among hospitalized patients worldwide included two studies from Ethiopia. The pooled estimates for colonization were 45.6% for ESBL-PE and 16.19% for CPE. [13]. In addition, existing studies of colonization by ESBL-PE and CPE in Africa revealed that colonization rates are high in the community, and at admission to hospital, colonization rates are 18% and 32%, respectively [10]. The prevalence of ESBL-PE in Ethiopia, based on various specimens, ranges from 18% to 49% [14,15], and the prevalence of CPE is 5.44% [16].

Gastrointestinal colonization with ESBL-PE and CPE may contribute to these extremely increasing infections. Although preventing ESBL-PE and CPE colonization is an interesting strategy for reducing drug-resistant infections, the existing review in Ethiopia focused on infections from different clinical specimens, such as blood, urine and discharge [14,15]. This systematic review and meta-analysis aimed to estimate the pooled prevalence of colonization by ESBL-PE and CPE in Ethiopia, using fecal specimens or rectal swabs.

## Methods

### Study design and protocol registration

This systematic review and meta-analysis followed a pre-registered protocol (PROSPERO ID: CRD42024550137) and adhered to the Preferred Reporting Items for Systematic Reviews and Meta-Analyses (PRISMA) 2020 checklist (S1 Table). In accordance with PRISMA 2020 item 24c, we report the following deviations from the original protocol:

1. **Context modification**: The initial plan focused on Africa; however, as a prior systematic review analyzed hospitalized patients globally with continent-based stratification, we revised the scope to include all patients in Ethiopia, removing restrictions on study settings.

2. **Amendment of participants/population**: Originally, only hospitalized patients were considered. Due to the limited availability of studies on this population, we expanded the inclusion criteria to encompass non-hospitalized patients, enhancing the robustness of our findings.

3. **Subgroup analysis modification**: Initially, studies were to be stratified by strain. However, given the insufficient number of studies per strain and the limited relevance of strain-based subgroup analysis; especially since pooled prevalence for each strain is already provided (S3 Fig). We instead stratified by setting, age, and outcome detection/confirmation methods while omitting strain-based analysis.

### Search strategy

Potential studies were identified using electronic databases including PubMed, Hinari, and Google Scholar on May 04/2024. The reference lists of identified studies were systematically searched for relevant studies. Articles were identified using MeSH terms and keywords of the title by using Boolean operators (OR and AND); for instance, we searched PubMed as follows. ("gastrointestine infections"[MeSH terms]) OR "asymptomatic infections"[MeSH terms]) AND "drug resistance, microbial"[MeSH terms] OR "drug resistance, microbial"[MeSH

terms] OR "antibiotic susceptibility"[MeSH terms]) OR "beta lactama*"[MeSH terms] OR ("extended-spectrum"[All Fields] AND "beta lactamases"[MeSH terms])) AND ("gram-negative bacteria [MeSH terms] OR "bacilli"[All Fields]) OR ("enterobacteriaceae"[MeSH terms] OR ("enterobacterales"[MeSH terms] OR "escherichia coli"[MeSH terms] OR ("klebsiella"[MeSH terms]) AND ("human*"[MeSH terms] OR "child*"[MeSH terms] OR "neona*"[MeSH terms] OR "adult*"[MeSH terms] OR "patient*"[MeSH terms]) AND "ethiopia"[MeSH terms] (S2 Table).

## The eligibility criteria

We applied the CoCoPop (Condition, Context, and Population) approach to determine the inclusion and exclusion criteria. In this framework, the prevalence of ESBL-PE and/or CPE was defined as the condition (CO), individuals providing fecal specimens or rectal swabs represented the population (POP), and Ethiopia was designated as the context (CO).

**Inclusion criteria and exclusion criteria.** We included all laboratory-based studies conducted in Ethiopia have reported the presence of ESBL-PE and/or CPE in human fecal specimens or rectal swabs, without restrictions on the year of publication. In addition, all participants from all age groups and full-text articles written in the English language were included in this review. However, case reports, case series studies, and specimens other than fecal specimens or rectal/perirectal swabs were excluded. Additionally, we excluded reports on multiple clinical specimens with fecal specimens for which it was difficult to determine the intestinal colonization rate.

## Study population

The participants were all of all age groups and had different sexes and health statuses and lived in different regions of Ethiopia.

## Outcome

The main outcomes were the colonization rate of ESBL-PE and CPE from human fecal specimens or rectal swabs in Ethiopia.

## Data extraction and management

All the records from different electronic databases were combined and properly exported to Endnote version 9.2. The articles were merged into one folder to identify and remove duplicate articles. All duplicate studies were removed, and the full-text articles were downloaded manually using Endnote software. Data screening was carried out twice by two reviewers (MT. and MA.), who independently reviewed the titles and abstracts of all relevant studies from the downloaded articles based on the predefined inclusion criteria.

All important parameters were extracted from each study by three authors (MT. MA. & GG.) independently by using Microsoft Excel 2019. Discrepancies between those authors were resolved through discussion with the other authors. The data extraction format was prepared according to the PRISMA guidelines. For each study, the primary author, year of publication, sample size, study design, study year, study area, study setting, age group, and methods of detection for ESBL and carbapenemase were extracted. In addition, the rates of total ESBL-PE colonization, total CPE colonization, and individual ESBL- and carbapenemase-producing strains were extracted.

## Quality assessment

The quality of studies was assessed using standard critical appraisal tools prepared by the Joanna Briggs Institute (JBI) for prevalence and cohort studies [17]. The quality of the included studies was assessed by two reviewers (MT. and MA.), independently. Discrepancies were resolved through discussion, and unresolved differences were resolved by consulting a third reviewer (GG). The quality assessment tool had 9 questions for prevalence studies and 11 questions for cohort studies. The overall scores ranged from 0–4, 5–6, and 7–9 for prevalence studies and from 0–4, 5–8, and 9–11 for cohort studies, which were declared to have high, moderate, and low risk of bias, respectively [18]. Finally, studies with a score of moderate or low risk of bias were included in the systematic review and meta-analysis (S3 Table).

## Statistical analysis

The extracted data were exported to STATA software version 11 for analysis. The pooled colonization rates of ESBL-PEs and CPEs were determined separately via a random effects model of DerSimonian-Laird method. Heterogeneity in meta-analyses is mostly inevitable due to differences in study quality, sample size, and method used across studies. The magnitude of heterogeneity between the included studies was quantitatively measured by the index of heterogeneity ($I^2$ statistics). An $I^2 < 25\%$ indicated low heterogeneity, an $I^2 = 50\text{-}75\%$ indicated moderate heterogeneity, and an $I^2 > 75\%$ indicated high heterogeneity. The significance of heterogeneity was determined by the p value of the $I^2$ statistic, and a p value less than 0.05 was considered evidence of heterogeneity [19].

To minimize the variance of estimated points between primary studies, a subgroup analysis was performed in reference to the study regions, study setting, age categories, and methods of confirmation for the outcome. Additionally, a sensitivity analysis was conducted to determine the influence of single studies on the pooled estimates. Publication bias and small study effects were checked by using a funnel plot test and Egger's test. A statistically significant Egger's test (p value < 0.05) indicated the presence of a small study effect [20].

## Ethical approval

Not needed.

# Results

## Study selection and identification

We identified a total of 527 articles from the available scientific databases. Of these, 392 were removed because they were duplications. One hundred thirty-five studies were screened by reading their titles and abstracts, and 93 studies were excluded because the outcome of interest related to this review was irrelevant. Additionally, 24 articles were removed because the specimen types were not appropriate. Three studies were excluded because the outcomes were not reported or because of a lack of full text. Finally, 15 studies were eligible and included in the final meta-analysis, as presented in the PRISMA flow diagram (Fig 1).

## Summary of the risk of bias results

Regarding the studies that met the quality criteria, 11 out of 15 (73.3%) had a low risk of bias, while 4 (26.7%) had a moderate risk of bias. No studies were excluded due to a high risk of bias (S3 Table).

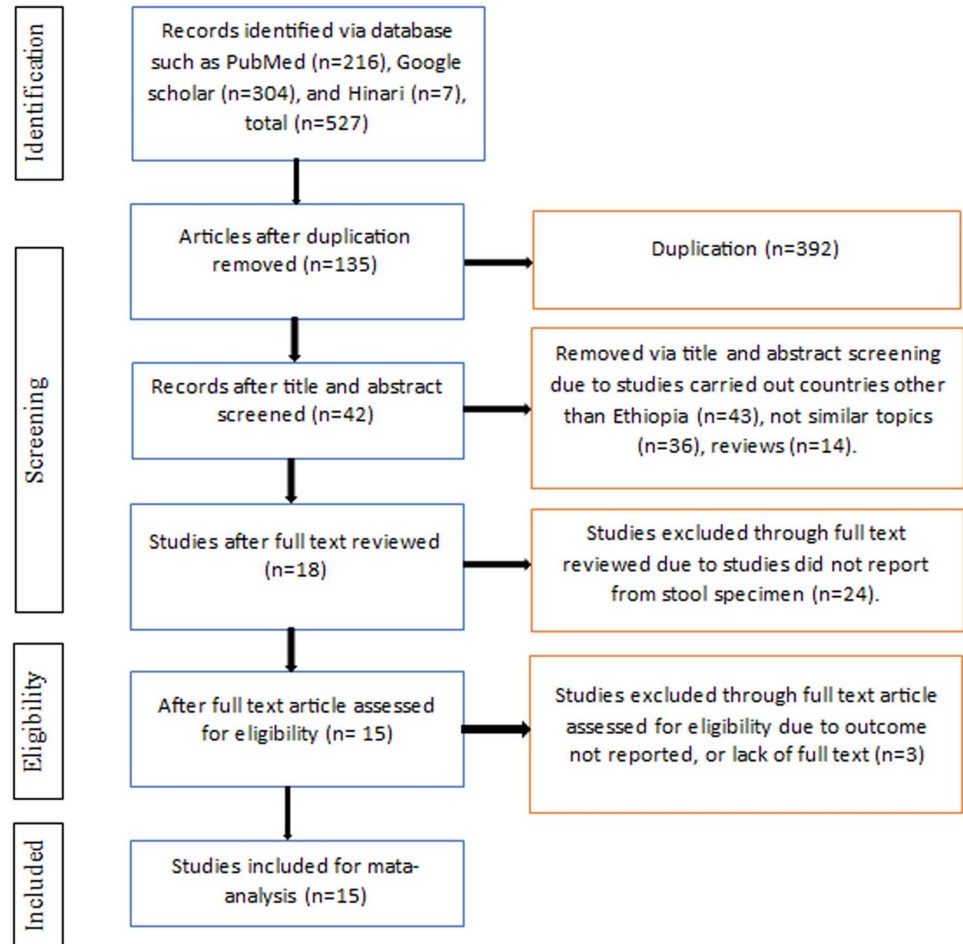

**Fig 1. A flow diagram of the study selection process for the meta-analysis of the rates of ESBL-PE and CPE colonization in Ethiopia, 2024.**

## Characteristics of the studies included in the systematic review and meta-analysis

Among the included studies, 11 (73.3%) studies were conducted and published after 2020. All of the included studies were cross-sectional surveys, except for one prospective cohort study. Thirteen (86.7%) studies were hospital- or health facility-based, whereas the remaining two studies were institutional-based and involved students' cafeteria. Of the 15 studies, 6 reported the ESBL-PE colonization rate, only 1 reported the CPE carriage rate, and 8 included both the ESBL-PE and CPE studies. All of the studies included in this review were from three regions and one city administration (Addis Ababa) (Table 1).

A total of 4329 study participants from 14 studies were included in the ESBL-PE meta-analysis. Out of 14 studies, only two detected ESBL-PE by using vitek-2, but others used the combination disk test (CDT) or double disk synergy test (DDST). The minimum and maximum sample sizes were 161 [21] and 476 [22], respectively. A minimum (6.3%) colonization rate of ESBL-PE was recorded from a single study conducted in Addis Ababa and Debre Berhan in the Amhara region [22], and a maximum (78.0%) colonization rate of ESBL-PE was reported from a study conducted in the Oromia region [23] (Table 1).

**Table 1. Characteristics of the individual studies included in the meta-analysis of ESBL-PE in Ethiopia, 2024.**

| Study (Author, Year) | Study year | Study area (Region) | Study design | Study population (Setting) | Age group | Sample size | Methods* | No of cases | Total ESBL-PE | ESBL-E. coli | ESBL-Klebsiella spp. | ESBL-Others$ | MDR among ESBL-PE |
|---|---|---|---|---|---|---|---|---|---|---|---|---|---|
| Temsegen et al, 2023 [23] | 2021 | Oromia | CS | Oncologic and nononcologic patient (H) | All age | 214 | CDT | 167 (78.0%) | 205 (95.8%) | 150 (70.1%) | 44 (20.6%) | 11 (5.1%) | 113 (55.1%) |
| Tola et al, 2021 [25] | 2017 | Addis Ababa | CS | Outpatient children (H) | Children | 269 | Vitek-2 | 46 (17.0%) | 47 (17.5%) | 39 (14.5%) | 8 (3%) | NR | 44 (93.6%) |
| Zakir et al, 2021 [26] | 2021 | South Ethiopia | CS | Hospitalized patient (H) | Neonates | 212 | DDST | 70 (33.0%) | 70 (33%) | 23 (10.8%) | 31 (14.6%) | 16 (7.5%) | 61 (87.1%) |
| Zenebe et al, 2023 [22] | 2020-2021 | Addis Ababa, and Amhara (T) | CS | Diarrheic and nondiarrheic (H) | Children | 476 | CDT | 30 (6.3%) | 30 (6.3%) | 30 (6.3%) | NR | NR | 30 (100%) |
| Shenkute et al, 2022 [27] | 2020-2021 | Amhara | CS | Hospitalized patient (H) | All age | 383 | CDT | 164 (42.8%) | 164 (42.8%) | 90 (23.5%) | 64 (16.7%) | 10 (2.6%) | NR |
| Kiros et al, 2023 [24] | 2022-2023 | Amhara | CS | Hospitalized patient (H) | All age | 383 | DDST | 102 (26.6%) | 102 (26.6%) | 58 (15.1%) | 40 (10.4%) | 4 (1.0%) | NR |
| Amare et al, 2022 [28] | 2021 | Amhara | CS | Apparently health food handler from students cafeteria (C) | Adult | 290 | CDT | 63 (21.7%) | 63 (21.7%) | 43 (14.8%) | 17 (5.9%) | 3 1.0% | NR |
| Diriba et al, 2020 [29] | 2018-2019 | South Ethiopia | CS | Apparently health food handler from students cafeteria (C) | Adult | 220 | DDST | 37 (16.8%) | 37 (16.8%) | NR | 37 (16.8%) | NR | NR |
| Aklilu et al, 2022 [30] | 2018-2019 | South Ethiopia | CS | Hospitalized patient (H) | All age | 421 | DDST | 146 (34.7) | 146 (34.7%) | 62 (14.7%) | 60 (14.3%) | 18 (4.3%) | 99 (70.7%) |
| Bayleyegn et al, 2020 [21] | 2021 | Amhara | CS | HIV positive outpatient (H) | Children | 161 | CDT | 32 (19.9%) | 32 (19.9%) | 26 (16.1%) | 6 (3.7%) | NR | 13 (40.6%) |
| Worku et al, 2022 [31] | 2019 | Amhara | CS | GIT complain patient (H) | All age | 384 | CDT | 66 (17.2%) | 66 (17.2%) | 41 (10.7%) | 22 (5.7%) | 3 (0.8%) | NR |
| Desta et al, 2016 [32] | 2012 | Addis Ababa | CS | Hospitalized patient (H) | All age | 267 | Vitek-2 | 139 (52.1%) | 151 (56.6%) | 106 (39.7%) | 45 (16.9%) | NR | NR |
| Wolde et al, 2024 [33] | 2021-2022 | Addis Ababa, and South Ethiopia (T) | CS | Hospitalized patient (H) | All age | 260 | DDST | 22 (8.5%) | 22 (8.5%) | 22 (8.5%) | NR | NR | NR |
| Amsalu et al, 2024 [34] | 2022 | Amhara | C | Pregnant woman and their neonates after 6 days of delivery (H) | Adult and neonates | 389 | DDST | 86 (22.1%) | 102 (26.2%) | 82 (21.1%) | 13 (3.3%) | 7 (1.8%) | NR |

Note: CS: cross-sectional study, C: cohort study, CDT: combination disk test, DDST: double disk synergy test, H: hospital, C: community, T: two regions, NR: no reported, No of cases: number of cases, ESBL-PE: extended-spectrum β-lactamase-producing *Enterobacterales*, Methods*: methods of confirmation for extended-spectrum β-lactamase, others$: *Proteus* spp. *Citrobacter* spp. and *Enterobacter* spp.

**Table 2. Characteristics of the individual studies included in the meta-analysis of CPE in Ethiopia in 2024.**

| Study (Author, Year) | Study year | Study area | Study design | Study population | Age group | Sample size | Method** | No of cases | Total CPE | CP-E. coli | CP-Klebsiella spp. | CPE-Others$$ |
|---|---|---|---|---|---|---|---|---|---|---|---|---|
| Temsegen et al, 2023 [23] | 2021 | Oromia | CS | Oncologic and nononco-logic patient (H) | All age | 214 | mCIM | 16 (7.5%) | 16 (7.5%) | 8 (3.7%) | 6 (2.8%) | 2 (0.9%) |
| Zakir et al, 2021 [26] | 2021 | South Ethiopia | CS | Neonates in intensive care units (H) | Neonates | 212 | mCIM | 5 (2.4%) | 5 (2.4%) | NR | 5 (2.4%) | NR |
| Zenebe et al, 2023 [22] | 2020-2021 | Addis Ababa, and Amhara (T) | CS | Diarrheic and nondiarrheic (H) | Children | 476 | mCIM | 4(0.8%) | 4(0.8%) | 4(0.8%) | NR | NR |
| Mekonnen et al, 2023 [35] | 2022 | Addis Ababa | CS | Hospitalized patient (H) | Adult | 384 | mCIM | 28 (7.3%) | 28 (7.3%) | 12 (3.1%) | 12 (3.1%) | 4 (1.1%) |
| Kiros et al, 2023 [24] | 2022 -2023 | Amhara | CS | Hospitalized patient (H) | All age | 383 | mCIM | 52 (13.6%) | 52 (13.6%) | 31 (8.1%) | 18 (4.7%) | 3 (0.8%) |
| Amare et al, 2022 [28] | 2021 | Amhara | CS | Apparently health food handler from students cafeteria (C) | Adult | 290 | mCIM | 7 (2.4%) | 7 (2.4%) | 4 (1.4%) | 3 (1.1%) | NR |
| Aklilu et al, 2022 [30] | 2018-2019 | South Ethiopia | CS | Hospitalized patient (H) | All age | 421 | mCIM | 6 (1.4%) | 6 (1.4%) | NR | 6 (1.4%) | NR |
| Worku et al, 2022 [31] | 2019 | Amhara | CS | GIT complain patient (H) | All age | 384 | mCIM | 4 (1.1%) | 4 (1.1%) | 1(0.3%) | 1(0.3%) | 2 (0.5%) |
| Desta et al, 2016 [32] | 2012 | Addis Ababa | CS | Hospitalized patient (H) | All age | 267 | Vitek-2 | 5 (1.9%) | 5 (1.9%) | 3 (1.1%) | 2 (0.7%) | NR |

Note: CS: cross-sectional study, No of cases: number of cases, NR: not reported, H: hospital, C: community, mCIM: modified carbapenem inactivation methods, T: Two regions, NR, no report, CPE: carbapenemase-producing *Enterobacterales*, CP: carbapenemase-producing, Methods** : methods of confirmation for carbapenemase, others$$: *Proteus* spp, *Citerobacter* spp. and *Enterobacter* spp.

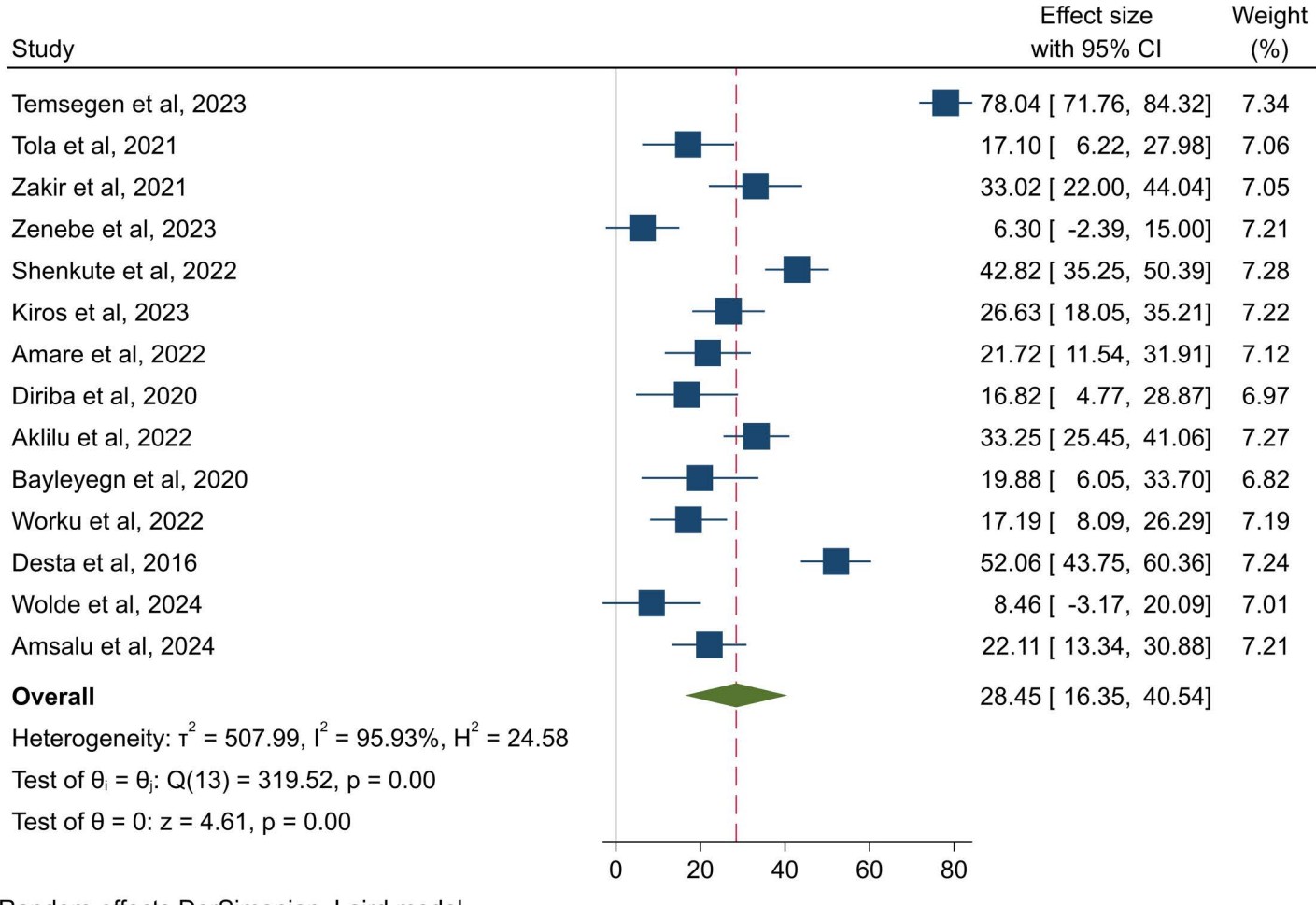

**Fig 2. Forest plot for the pooled colonization rate of ESBL-PE in Ethiopia.**

Furthermore, the CPE meta-analysis was performed on 9 studies with 3031 study participants. The minimum and maximum colonization rates of CPE were reported by Zenebe et al. (0.8%) [22] and Kiros et al. (13.6) [24], respectively ([Table 2]).

## Colonization with extended-spectrum β-lactamase-producing Enterobacterales in Ethiopia

The overall pooled colonization rate of ESBL-PE was 28.5% (95% CI: 16.4–40.5%), with a high level of heterogeneity ($I^2$ = 95.9%, p < 0.001), as presented in ([Fig 2]). The majority of ESBL-PE were *E. coli* (20.6%, 95% CI: 9.3–31.9%, $I^2$ = 94.4%, p < 0.001), followed by *Klebsiella* spp. (11.1%, 95% CI: 7.7–14.6%, $I^2$ = 20.2%, p = 0.245) ([S4 Figs]). The predominant species among the *Klebsiella* spp. were *K. pneumoniae* (9.1%, 95% CI: 6.0–12.2%, $I^2$ = 0.0%, p = 0.644) and *K. oxytoca* (3.6%, 95% CI: -1.1-8.3%, $I^2$ = 0.0%, p = 0.895), as indicated in ([S4 Figs]).

## Colonization with carbapenemase-producing Enterobacterales in Ethiopia

Only 9 studies were reported for carbapenemase-producing isolates, and the pooled estimate of CPE colonization was 4.4% (95% CI: 0.9–7.9%, $I^2$ = 0.0%, p = 0.64), with a non-significant

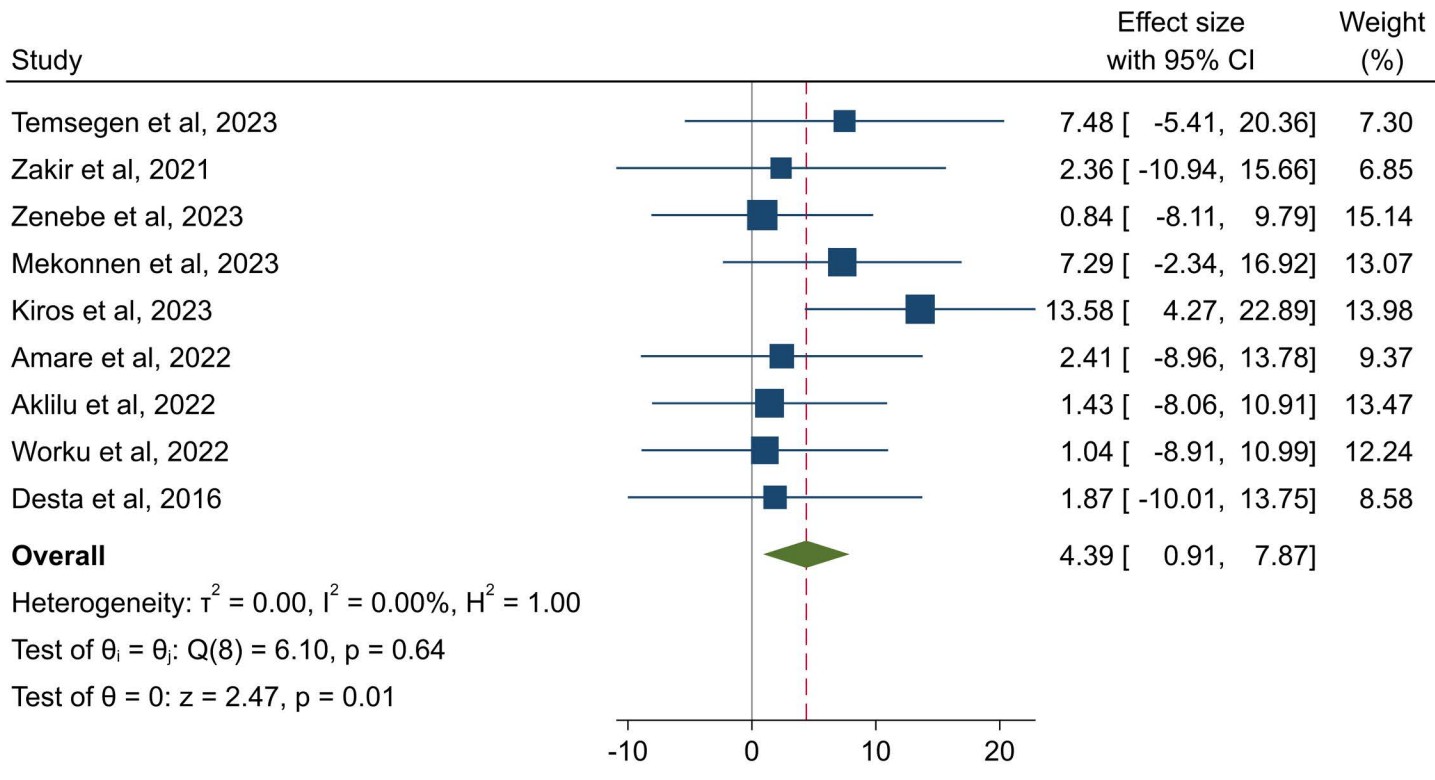

**Fig 3. Forest plot for the rate of CPE colonization rate in Ethiopia.**

low level of heterogeneity ($I^2$ = 0.0%), as presented in (Fig 3). The predominant CPE was *E. coli* at 2.7% (95% CI: -1.3–6.7%, $I^2$ = 0.0%, p = 0.941), followed by *Klebsiella* spp. at 2.1% (95% CI: -1.7–5.9%, $I^2$ = 0.0%, p = 0.999), *K. pneumoniae* at 2.1% (95% CI: -2.0–6.3%, $I^2$ = 0.0%, p = 0.999), and *K. oxytoca* at 0.6% (95% CI: -5.1-6.3%, $I^2$ = 0.0%, p = 1.000), as indicated in (S4 Figs).

## Publication bias

Publication bias was assessed to determine whether the selected studies represented the original population or were influenced by bias related to published and unpublished studies. The ESBL-PE analysis revealed an asymmetric funnel plot, indicating the presence of publication bias (Fig 4). Similarly, Egger's test showed significant publication bias (p = 0.004), as indicated in Table 3. However, the analysis of CPE showed a symmetric funnel plot, with an equal visual distribution of study effect sizes on the funnel plot (Fig 5). Additionally, Egger's test revealed non-significant publication bias (p = 0.772), indicating the absence of publication bias, as presented in Table 3.

## Trim-and-fill analysis of the pooled colonization rate of ESBL-PE in Ethiopia

A trim-and-fill analysis was conducted to address the observed publication bias. After imputing data on the left, the pooled colonization rate of ESBL-PE in Ethiopia remained stable at 28.5% (95% CI: 16.4–40.5%) (Table 4). Conversely, when analyzing 17 studies with three data points imputed on the right, the pooled colonization rate was slightly higher at 33.3% (95% CI: 22.43–44.15%) (Table 5).

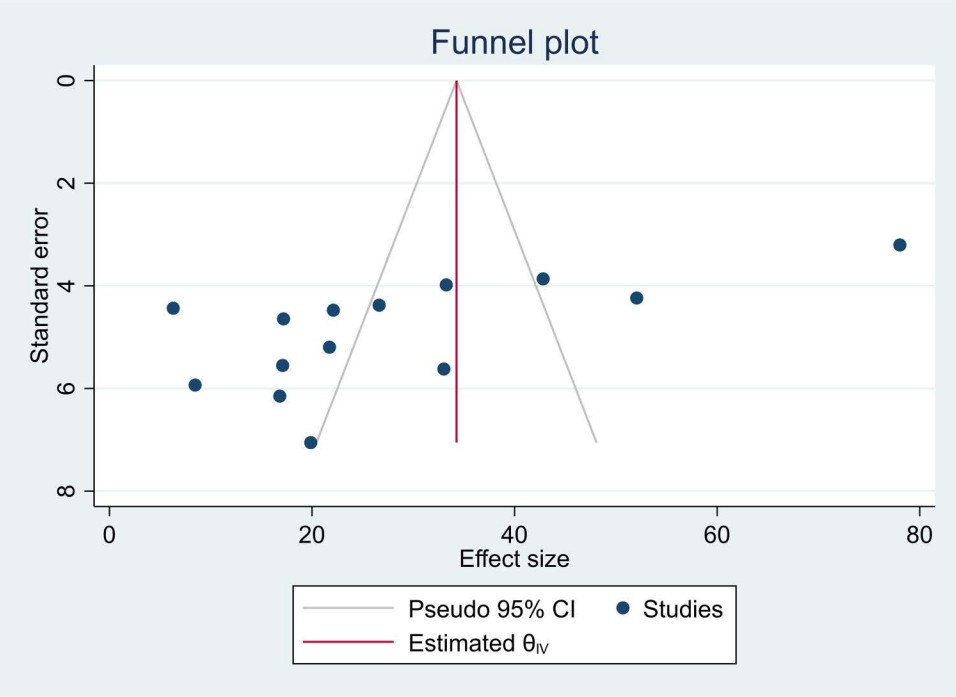

**Fig 4. Funnel plot showing publication biases of the colonization rate of ESBL-PEs in Ethiopia.**

**Table 3. Egger's test for the pooled estimates of the colonization rates of ESBL-PE and CPE.**

| Category | Number of studies | Pooled estimate (95% CI) | Heterogeneity | | Eggers regression test | |
|---|---|---|---|---|---|---|
| | | | I2 | P value | P value | 95% CI |
| ESBL-PE colonization | 14 | 28.5% (16.4%–40.5%) | 95.9% | <0.001 | 0.004 | -27.34–6.37 |
| CPE colonization | 9 | 4.4% (0.9%–7.9%) | 0.0% | 0.64 | 0.772 | -6.08–4.71 |

Note: ESBL-PE: extended-spectrum β-lactamase-producing *Enterobacterales,* CPE: carbapenemase-producing *Enterobacterales*

## Sensitivity analysis

Sensitivity analysis was carried out to detect any potential outlier studies. Based on the random effects model, no single study had a disproportionate impact on the overall pooled estimates of ESBL-PE and CPE colonization rates. The findings revealed that the estimates from all included studies were within the pooled estimate's confidence interval, indicating the reliability of the aggregated results (**Table 6**).

## Subgroup analysis of the ESBL-PE and CPE colonization rates

Subgroup analysis of ESBL-PE was carried out across the administrative regions of the country, study setting, age group and methods of detecting ESBL-PE. Therefore, the highest pooled estimate of ESBL-PEs (78%) was observed in the Oromia regional state, followed by Addis Ababa (34.8%), whereas the lowest pooled estimate was reported in two regions (7.1%). When subgroup analysis was performed by study setting, the pooled prevalence of fecal carriage was greater in the hospital (29.9%) than in the community (19.7%). Additionally, the highest colonization rate was reported by the authors who used the vitek-2 machine (34.8%) and

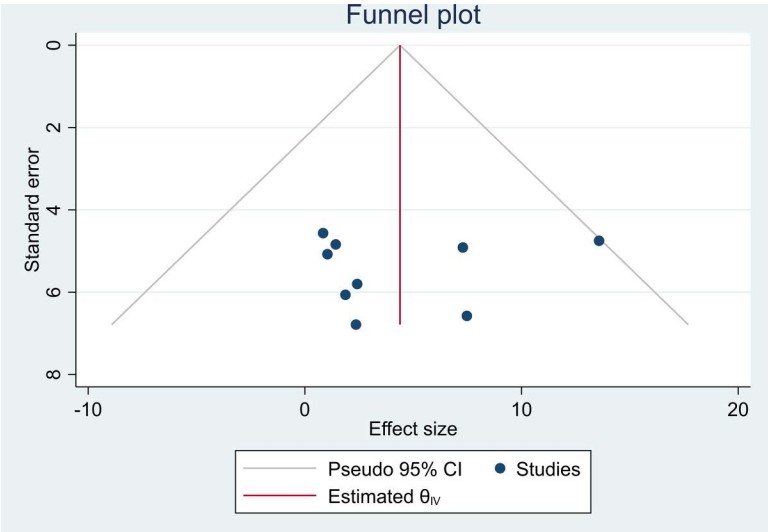

**Fig 5. Funnel plot showing publication biases related to the colonization rate of CPE in Ethiopia.**

**Table 4. Nonparametric trim-and-fill analysis of publication bias imputing on the left.**

| Nonparametric trim-and-fill analysis of publication bias | |
| --- | --- |
| **Linear estimator, imputing on the left** | |
| Iteration | |
| Parameter | Value |
| Number of studies | 14 |
| Model | Random-effects |
| Method | DerSimonian-Laird |
| Observed | 14 |
| Imputed | 0 |
| Pooling | |
| Parameter | Value |
| Model | Random-effects |
| Method | DerSimonian-Laird |
| Results | |
| Studies | Effect size (95% conf. interval) |
| Observed | 28.445 (16.353–40.537) |
| Observed + Imputed | 28.445 (16.353–40.537) |

CDT (31.2%), but the lowest pooled estimate was recorded by the DDST (23.9%) method, as indicated in (**Table 7**).

Similarly, subgroup analysis of CPE was performed by region of the country, study setting, and age group. The highest colonization rate of CPE (7.5%) was observed in Oromia, followed by the Amhara regional state (6.0%). Furthermore, the pooled fecal carriage rate of CPE was greater in the hospital (4.6%) than in the community (2.4%) (**Table 7**).

## Multidrug resistance profiles

The analysis was performed by using a random effects model for 6 studies that reported multidrug resistance (MDR) profiles among ESBL-PE isolates. The pooled

**Table 5. Nonparametric trim-and-fill analysis of publication bias imputing on the right.**

| Nonparametric trim-and-fill analysis of publication bias | |
| --- | --- |
| Linear estimator, imputing on the right | |
| Iteration | |
| Parameter | Value |
| Number of studies | 17 |
| Model | Random-effects |
| Method | DerSimonian-Laird |
| Observed | 14 |
| Imputed | 3 |
| Pooling | |
| Parameter | Value |
| Model | Random-effects |
| Method | DerSimonian-Laird |
| Results | |
| Studies | Effect size (95% conf. interval) |
| Observed | 28.445 (16.353–40.537) |
| Observed + Imputed | 33.287 (22.427–44.148) |

**Table 6. Sensitivity analysis of ESBL-PE and CPE colonization in Ethiopia, 2024.**

| Sensitivity analysis of ESBL-PE | | | Sensitivity analysis of CPE | | |
| --- | --- | --- | --- | --- | --- |
| Study omitted | Estimate | 95% Confidence Interval | Study omitted | Estimate | 95% Confidence Interval |
| Temsegen et al, 2023 | 24.68 | 17.03-32.34 | Temsegen et al, 2023 | 4.12 | 0.53-7.76 |
| Tola et al, 2021 | 29.30 | 16.59-42.01 | Zakir et al, 2021 | 4.54 | 0.93-8.15 |
| Zakir et al, 2021 | 28.09 | 15.18-40.99 | Zenebe et al, 2023 | 5.03 | 1.25-8.80 |
| Zenebe et al, 2023 | 30.18 | 17.97-42.39 | Mekonnen et al, 2023 | 3.96 | 0.22-7.67 |
| Shenkute et al, 2022 | 27.30 | 14.12-40.47 | Kiros et al, 2023 | 2.89 | -0.85-6.65 |
| Kiros et al, 2023 | 28.57 | 15.51-41.63 | Amare et al, 2022 | 4.59 | 0.94-8.25 |
| Amare et al, 2022 | 28.98 | 16.11-41.79 | Aklilu et al, 2022 | 4.85 | 1.11-8.59 |
| Diriba et al, 2020 | 29.31 | 16.64-41.99 | Worku et al, 2022 | 4.86 | 1.14-8.57 |
| Aklilu et al, 2022 | 28.05 | 14.80-41.29 | Desta et al, 2016 | 4.63 | 0.99-8.27 |
| Bayleyegn et al, 2020 | 29.07 | 16.39-41.75 | Combined | 4.39 | 0.91-7.87 |
| Worku et al, 2022 | 29.31 | 16.54-42.07 | – | – | – |
| Desta et al, 2016 | 26.60 | 13.83-39.36 | – | – | – |
| Wolde et al, 2024 | 29.95 | 17.49-42.41 | – | – | – |
| Amsalu et al, 2024 | 28.92 | 15.98-41.87 | – | – | – |
| Combined | 28.45 | 16.35-40.54 | – | – | – |

Note: ESBL-PE: extended-spectrum β-lactamase-producing *Enterobacterales,* CPE: carbapenemase-producing *Enterobacterales*

estimate of MDR was 71.7% (95% CI: 55.25–88.05, I² = 92.9%, p < 0.001), as indicated in (**Fig 6**).

## Discussion

Extended-spectrum β-lactamase- and carbapenemase-producing *Enterobacterales* are major global threats due to their high burden and ability to spread resistance to other bacteria [3]. In our systematic review and meta-analysis, the overall pooled colonization rates of ESBL-PE

**Table 7. Subgroup analysis of ESBL-PE and CPE colonization rates in Ethiopia, 2024.**

| Characteristics | No. of Studies | Sample size | No. of cases | Estimated pooled colonization rate of ESBL-PE (95% CI) | Heterogeneity | |
|---|---|---|---|---|---|---|
| | | | | | I² (%) | P value |
| Subgroup analysis of ESBL-PE | | | | | | |
| **Region** | | | | | | |
| Addis Ababa | 2 | 536 | 185 | 34.76 (0.51, 69.02) | 96.0 | <0.001 |
| Oromia | 1 | 214 | 167 | 78.04 (71.76, 84.32) | NA | NA |
| South Ethiopia | 3 | 853 | 253 | 28.38 (18.60, 38.17) | 63.9 | 0.063 |
| Two region | 2 | 736 | 52 | 7.08 (0.11, 14.04) | 0.0 | 0.771 |
| Amhara | 6 | 1990 | 513 | 25.46 (17.00-33.93) | 79.6 | <0.001 |
| **Study setting** | | | | | | |
| Hospital | 12 | 3819 | 1070 | 29.93 (16.43-43.43) | 96.4 | <0.001 |
| Community | 2 | 510 | 100 | 19.68 (11.90-27.46) | 0.0 | 0.542 |
| **Publication year** | | | | | | |
| 2016-2020 | 3 | 648 | 208 | 29.98 (5.16-54.81) | 93.2 | <0.001 |
| 2021-2024 | 11 | 3681 | 962 | 28.03 (13.76-42.30) | 96.5 | <0.001 |
| **Methods of ESBL-detection** | | | | | | |
| CDT | 6 | 1908 | 522 | 31.15 (6.33-55.97) | 97.9 | <0.001 |
| Vitek-2 | 2 | 536 | 185 | 34.76 (0.51-69.02) | 96.0 | <0.001 |
| DDST | 6 | 1885 | 463 | 23.93 (16.71-31.15) | 69.6 | 0.006 |
| **Age group** | | | | | | |
| All age | 7 | 2312 | 806 | 37.13 (18.88-55.38) | 97.1 | <0.001 |
| Children | 3 | 906 | 108 | 13.33 (4.72-21.93) | 46.4 | 0.155 |
| Neonate | 2 | 601 | 156 | 27.03 (16.39-37.68) | 56.6 | 0.129 |
| Adult | 2 | 510 | 100 | 19.68 (11.90-27.46) | 0.0 | 0.542 |
| Subgroup analysis of CPE | | | | | | |
| **Region** | | | | | | |
| Addis Ababa | 2 | 651 | 33 | 5.14 (-2.34- 12.62) | 0.0 | <0.487 |
| Oromia | 1 | 214 | 16 | 7.48 (-5.41-20.36) | NA | NA |
| South Ethiopia | 2 | 633 | 11 | 1.74 (-5.98-9.46) | 0.0 | 0.911 |
| Two region | 1 | 476 | 4 | 0.84 (-8.11- 9.79) | NA | NA |
| Amhara | 3 | 1057 | 63 | 6.01 (-2.16-14.17) | 48.3 | <0.145 |
| **Study setting** | | | | | | |
| Hospital | 8 | 2741 | 120 | 4.60 (0.94-8.25) | 0.0 | 0.543 |
| Community | 1 | 290 | 7 | 2.41 (8.96-13.78) | NA | NA |
| **Age group** | | | | | | |
| All age | 5 | 1669 | 83 | 5.22 (0.07-10.37) | 16.8 | 0.307 |
| Children | 1 | 476 | 4 | 0.84 (-8.11- 9.79) | NA | NA |
| Neonate@ | 1 | 212 | 5 | 2.36 (10.94-15.66) | NA | NA |
| Adult | 2 | 674 | 35 | 5.25 (-2.09- 12.60) | 0.0 | 0.521 |

Note: ESBL-PE: extended-spectrum β-lactamase-producing *Enterobacterales,* CPE: carbapenemase-producing *Enterobacterales,* NA: not applicable, @: some isolates from pregnant women included  two regions: studies conducted from both the Addis Ababa and Amhara regions

and CPE in Ethiopia were 28.5% (95% CI: 16.4%-40.5%) and 4.4% (95% CI: 0.9%–7.9%), respectively. The prevalence of ESBL-PE colonization in this study is consistent with reports from two meta-analysis studies in sub-Saharan Africa on admitted patients and children with a similar pooled prevalence of 32% [10,36]. Moreover, the findings of this study are comparable to those of two systematic reviews conducted on residents of long-term care facilities and

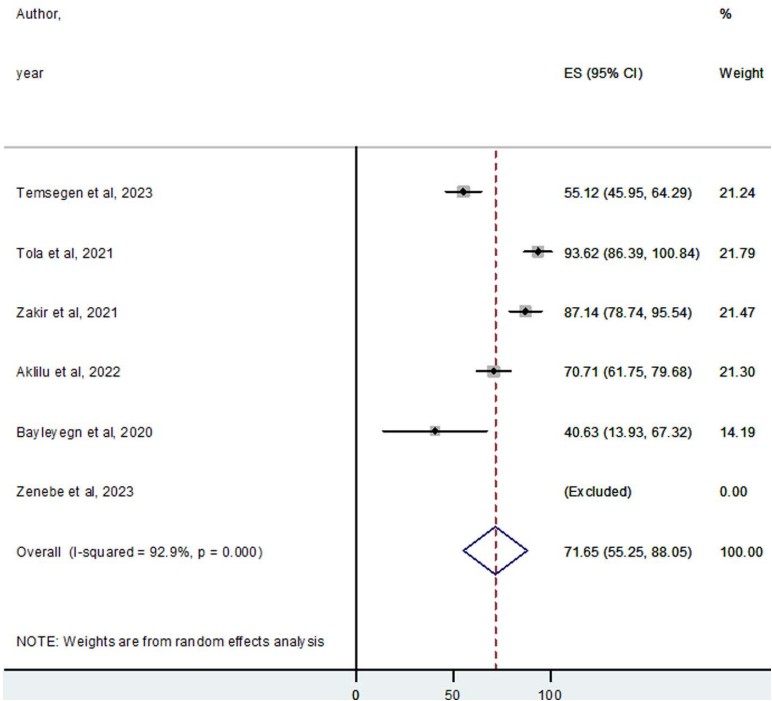

**Fig 6. Forest plot for MDR profiles among ESBL-PE isolates in Ethiopia, 2024.**

among patients with hematological malignancies, in which the percentages of patients with pooled ESBL-PE colonization were 18% and 19%, respectively [9,37].

This higher colonization rate of ESBL-PE in our and other meta-analyses indicates that ESBL-PE could be disseminated among healthcare institutions, communities and even across the country in Africa [38]. Moreover, in the present meta-analysis, the high ESBL-PE and CPE carriage rates might be related to the high consumption of antibiotics, including beta-lactam antibiotics, in Ethiopia, leading to the emergence and spread of plasmid-mediated beta-lactamase genes [39]. Another suggested reason is the low control mechanism for antibiotic usage, which causes overuse and misuse of antibiotics in healthcare facilities and animal hus-bandry in Ethiopia [11,40].

However, the colonization rates of ESBL-PE and CPE in the current meta-analysis are lower than those reported in a meta-analysis of hospitalized patients at the global level, with pooled estimates of ESBL-PE and CPE of 45.6% and 16.2%, respectively [13], and higher than those reported in a meta-analysis of healthy individuals in a community with pooled ESBL-PE carriage of 14% [8]. This variation might be due to geographical location, the study popula-tion, the study period and the methods used for screening and confirmation of ESBL-PE and CPE.

According to the present review, the heterogeneity between studies was high ($I^2 = 95.9\%$, p < 0.001) for the colonization rate of ESBL-PE. Therefore, subgroup meta-analysis was carried out by region, and the highest percentages of patients with ESBL-PE (78%) and CPE (7.5%) were recorded in the Oromia region [23]. The number of studies can affect the pooled esti-mates: in the Oromia regional state, a single study was involved in this analysis.

Moreover, this study was conducted in oncologic and nononcologic patients. Therefore, oncologic patients are more prone to be colonized with ESBL-PE and CPE due to frequent

contact with hospitals, which increases the horizontal spread of resistant bacteria [37]. In addition, frequent exposure to antibiotics leads to the selection of ESBL-PE and CPE in the gastrointestinal tract of patients [41]. The lowest pooled colonization rates of ESBL-PE (7.1%) and CPE (0.8%) were reported from a single study conducted in the Addis Ababa and Amhara regions [22]. The lowest carriage rate may be because the authors focused on a single bacterium, such as *E. coli.* The prevalence of total *Enterobacterales* can be reduced, as these types of studies excluded other species.

According to the subgroup analysis by study setting, the pooled prevalence of fecal ESBL-PE and CPE was greater in the hospital (29.9% and 4.6%) than in the community (19.7% and 2.4%). This higher colonization rate from hospitals might be due to the selection of resistant bacteria in the intestine, as they use antibiotics during hospitalization or acquire resistant strains from hospitals [41]. This is supported by Ruef et al., who reported that the number of patients who acquired resistant strains after two days of hospitalization was double the prevalence at the first admission [36].

Furthermore, a subgroup analysis based on methods of ESBL-PE detection (confirmatory tests) revealed that the highest colonization rate was reported by authors who used the VITEK-2 machine (34.8%), followed by the CDT method (31.2%). In comparison, the lowest pooled estimate was recorded using the DDST method (23.9%). This is due to the vitek-2 machine and CDT are more sensitive than DDST for the detection of ESBL-producing Enterobacterales [42, 43].

In the current meta-analysis, the pooled estimated MDR profile among the ESBL-PE isolates was 71.7% (95% CI: 55.25–88.05%, $I^2$ = 92.9%, p < 0.001). A poor antibiotic stewardship program and infection prevention mechanism may contribute to the high carriage of MDR strains among ESBL-PE in Ethiopia [11]. Additionally, these higher MDR profiles may be because ESBL-producing Enterobacterales are often resistant to several classes of antibiotics, as plasmids with genes encoding ESBL often carry other resistance determinants. Therefore, many ESBL-producing Enterobacterales are also resistant to non-beta-lactam antibiotics, including fluoroquinolones, aminoglycosides, trimethoprim, tetracycline, sulfonamides, and chloramphenicol [44, 45].

## Limitations

Some studies have focused on only single species, such as ESBL-producing *E. coli, and* these studies can underestimate the overall colonization rate.

## Conclusion and recommendations

Approximately one-quarter of Ethiopians are colonized with ESBL-PE, and about one in 25 with CPE, based on studies with moderate-to-low risk of bias. Results for ESBL-PE showed high heterogeneity. Subgroup analysis revealed the highest ESBL-PE prevalence in the Oromia region, with rates higher in hospitals than in communities. Detection using the VITEK-2 machine reported the highest prevalence, followed by CDT, while DDST showed the lowest.

The MDR profile among ESBL-PE isolates was significant. Colonization facilitates the spread of ESBL-PE and CPE and can lead to severe infections. To mitigate this, protecting the gut microbiome and strengthening efforts against antibiotic resistance are essential.

## Supporting information

**S1 Table.  PRISMA checklist.**
(DOCX)

**S2 Table. Example of searching strategy from PubMed.**
(DOCX)

**S3 Table. Quality assessment of the studies included in a systematic review and meta-analysis on the colonization rate of ESBL-PE and CPE.**
(DOCX)

**S4 Figs. Forest plots showed the pooled colonization rate of ESBL-PE and CPE by each bacterial species.**
(DOCX)

## Acknowledgments

The authors would like to thank the authors of each study. We also acknowledge the College of Medicine and Health Sciences, University of Gondar.

## Author contributions

**Conceptualization:** Mitkie Tigabie.

**Data curation:** Mitkie Tigabie, Muluneh Assefa.

**Formal analysis:** Mitkie Tigabie, Muluneh Assefa.

**Funding acquisition:** Mitkie Tigabie.

**Investigation:** Mitkie Tigabie.

**Methodology:** Mitkie Tigabie, Getu Girmay, Yalewayker Gashaw, Muluneh Assefa.

**Project administration:** Mitkie Tigabie.

**Resources:** Mitkie Tigabie.

**Software:** Mitkie Tigabie, Getu Girmay, Muluneh Assefa.

**Supervision:** Mitkie Tigabie, Getu Girmay, Yalewayker Gashaw, Getachew Bitew, Abebe Birhanu, Eden Getaneh, Azanaw Amare, Muluneh Assefa.

**Validation:** Mitkie Tigabie, Getachew Bitew, Abebe Birhanu, Eden Getaneh, Azanaw Amare.

**Visualization:** Mitkie Tigabie, Getu Girmay, Yalewayker Gashaw, Getachew Bitew, Abebe Birhanu, Eden Getaneh, Azanaw Amare.

**Writing – original draft:** Mitkie Tigabie.

**Writing – review & editing:** Mitkie Tigabie, Getu Girmay, Yalewayker Gashaw, Getachew Bitew, Abebe Birhanu, Eden Getaneh, Azanaw Amare, Muluneh Assefa.

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
