## [Decision Letter · Decision Letter 0]

27 Dec 2024

PONE-D-24-33532Colonization with extended-spectrum β-lactamase and carbapenemase-producing Enterobacterales in Ethiopia: a systematic review and meta-analysisPLOS ONE

Dear Dr. Tigabie,

Thank you for submitting your manuscript to PLOS ONE. After careful consideration, we feel that it has merit but does not fully meet PLOS ONE’s publication criteria as it currently stands. Therefore, we invite you to submit a revised version of the manuscript that addresses the points raised during the review process.

**I really appreciate this paper that explores an important topic. I apologize for the time this review has taken.**

**Thank you for responding to all the comments from reviewers 1 and 2. You don't necessarily have to make the requested corrections, but I encourage you to respond to all the comments. Thank you for carefully addressing the comments from the statistical expert.**

We look forward to receiving your revised manuscript.

Kind regards,

Lorenzo Righi

Academic Editor

PLOS ONE

**Journal Requirements:**

6. As required by our policy on Data Availability, please ensure your manuscript or supplementary information includes the following: 

Reviewers' comments:

Reviewer's Responses to Questions

**Comments to the Author**

1. Is the manuscript technically sound, and do the data support the conclusions?

Reviewer #1: Partly

Reviewer #2: Yes

Reviewer #3: Partly

2. Has the statistical analysis been performed appropriately and rigorously? 

Reviewer #1: I Don't Know

Reviewer #2: Yes

Reviewer #3: No

3. Have the authors made all data underlying the findings in their manuscript fully available?

Reviewer #1: Yes

Reviewer #2: Yes

Reviewer #3: Yes

4. Is the manuscript presented in an intelligible fashion and written in standard English?

Reviewer #1: No

Reviewer #2: Yes

Reviewer #3: Yes

5. Review Comments to the Author

**Reviewer #1:**  This is a systematic review and meta-analysis of colonisation rates of extended-spectrum beta-lactamase-producing and carbapenemase-producing enterobacterales in Ethiopia. These are bacterial infections that are resistant to common antibiotics and can spread in healthcare and non-healthcare settings. They are usually harmless to healthy colonised individuals, but they can be dangerous to vulnerable patients. The review seems to be conducted and reported well, but the writing could be improved in places and should be checked throughout. I have a few comments.

Abstract:

The conclusion in the abstract does not clearly represent the results of the meta-analysis, as it does not summarise the conclusion in the main paper. A summary could include something along the lines of “around a quarter of Ethiopians are colonised with ESBL-PE, and around one in 25 is colonised with CPE.” It should also make clear that it is based on a meta-analysis of studies judged to be of moderate-to-low risk of bias, and that the ESBL-PE study results were highly varied (high heterogeneity).

Introduction:

Line 82 – Should this be ESBL-PE rather than ESBLs?

Line 90 – It might be good to add that this is “in various specimens” here and add “using faecal specimens or swabs” to line 102.

Lines 100 to 116 – It would be good to explain, somewhere, why this has changed from the protocol, which specifies hospitalised patients in Africa.

Methods:

Line 116 – see SF1 for a completed checklist.

Line 118 - Are these examples (electronic databases such as) or the full list of databases? All the sources searched should be clearly reported here or in the supplement.

Line 136 – the “No” doesn’t make sense here – should it be “Non-laboratory-based studies that reported…”?

Line 156 – Was each record screened twice – once by each reviewer? How were differing decisions resolved? Or was each record screened by one reviewer?

Line 170 – This sentence doesn’t make sense should it be …tools had 9 and 11 questions…?

Lines 171 to 172 – This sentence is not needed as this is explained in the next sentence.

Lines 186 to 188 – It would be good to clearly report the changes to the protocol. The plan was for subgroups by region, strain and year of publication, so this adds setting, age and methods of outcome confirmation and removes strain and year of publication.

Results:

Lines 201 to 205 – There seems to be an error in the numbers reported here as 135 minus 96, 24 and 3 is 12 not 15. In the flow chart the 96 is 93.

Discussion:

Line 361 – This sentence needs to be rewritten: CDT (31.2%) was comparable, and the lowest pooled estimate was…

Lines 378 to 383 - The conclusion in the paper is appropriate, but it could be written more clearly. It should also make clear that it is based on studies judged to be of moderate-to-low risk of bias, and that the ESBL-PE estimate has high heterogeneity.

Figure 1. This indicates that all the 527 records were identified from PubMed, Google Scholar and Hinari – were there none from the other databases or were they not searched?

General:

Just a question, as I’m not a statistics expert, but is the probability reported the p-value for the heterogeneity or for the effect? For example, in Figure 3, the I-squared is zero, so no heterogeneity, the p-value is not significant (which is true for no significant heterogeneity), but the diamond doesn’t cross zero (the line of no effect) so should the p-value for the effect be significant?

Were there any studies that did not meet the quality criterion? It seems that all fifteen studies were at moderate or low risk of bias, so I assume that no studies were actually excluded due to a high risk of bias? If this is the case, it might be good to state it clearly at the beginning of the results section, with a summary of the risk of bias results.

I hope that my comments are helpful.

**Reviewer #2:**  1: A few places, the wording is unclear:

136-139

170

378-383

2: Line 202 and flow chart: There is a mismatch between the flow chart and the text. When doing the calculations, the flowchart match perfectly, by subtracting 93 (43 + 36 + 14) from 135, with 42 studies left. In the text, however, 93 has been replaced by 96. 96 should probably be 93, then everything fits perfectly.

**Reviewer #3: ** The systematic review appeared to be fairly comprehensive with search strategy, eligibility checks, main outcomes checked of the colonization rate of ESBL-PE and CPE from human fecal/rectal swab specimens in Ethiopia, data extraction and management outlines according to PRISMA and quality of study assessment using standard critical appraisal tools prepared by the Joanna Briggs Institute (JBI) for prevalence and cohort studies as is the case in this application. The data analysis endpoint was mainly colonization rate of ESBL-PEs and CPEs determined separately via a random effects model which is certainly appropriate in this context. Heterogeneity was assessed via the I square statistic. Sensitivity analysis was conducted to determine the influence of single studies on the pooled estimates. Publication bias and small study effects were checked by using a funnel plot test and Egger’s test. It appears that all the elements of this systematic review and meta-analysis were in place. The manuscript was well organized.

When evaluating the publication bias the authors note that the ESBL-PE analysis revealed an asymmetric funnel plot, which indicates the presence of publication bias (Fig. 4). Similarly, Egger’s test revealed significant publication bias (P=0.004), as indicated in Table 3. Also in Fig 5. the Funnel plot showed publication biases related to the colonization of CPE. The investigators should explain more completely, if possible, the sources of the asymmetry in both plots.

Also the interpretation of sensitivity is not detailed sufficiently for the reader in Table 4. In addition, in Table 5, the possible sources of heterogeneity, especially for those with an I squared above 90 and statistically significant (p<0.05) should be explained, if possible, for the reader.

The investigators use the statistic, ES, on the Forest plots throughout the manuscript and in supplemental file 3. Explain that it is the colonization rate. If otherwise, please define it. Also the manuscript should be edited to be sure the reader knows what the p-values are measuring, much like the authors did for explaining the p-values associated with heterogeneity and publication bias.

6. PLOS authors have the option to publish the peer review history of their article (what does this mean? ). If published, this will include your full peer review and any attached files.

**Do you want your identity to be public for this peer review?** For information about this choice, including consent withdrawal, please see our Privacy Policy .

Reviewer #1: **Yes: ** Claire Khouja

Reviewer #2: **Yes: ** Louise Schow Guski

Reviewer #3: No

---

## [Author Response · Author response to Decision Letter 1]

22 Jan 2025

Response to Reviewers

Authors: We appreciate for spending your precious time and forwarding your valuable comments, which have significantly improved our manuscript. We are also grateful for this positive feedback. Please see below, bold, for a point-by-point response to the reviewers. We've copied your comments and responses below to make things easier for you. All line numbers refer to the revised manuscript file.

Reviewer's Responses to Questions

Comments to the Author

1. Is the manuscript technically sound, and do the data support the conclusions?

Reviewer #1: Partly

Reviewer #2: Yes

Reviewer #3: Partly

2. Has the statistical analysis been performed appropriately and rigorously?

Reviewer #1: I Don't Know

Reviewer #2: Yes

Reviewer #3: No

3. Have the authors made all data underlying the findings in their manuscript fully available?

Reviewer #1: Yes

Reviewer #2: Yes

Reviewer #3: Yes

4. Is the manuscript presented in an intelligible fashion and written in standard English?

Reviewer #1: No

Reviewer #2: Yes

Reviewer #3: Yes

5. Review Comments to the Author

Authors: Thank you for your positive feedback; we appreciate your feedback. We have revised the entire manuscript as necessary and have attempted to address the comments from the reviewers.

Reviewer #1: This is a systematic review and meta-analysis of colonisation rates of extended-spectrum beta-lactamase-producing and carbapenemase-producing enterobacterales in Ethiopia. These are bacterial infections that are resistant to common antibiotics and can spread in healthcare and non-healthcare settings. They are usually harmless to healthy colonised individuals, but they can be dangerous to vulnerable patients. The review seems to be conducted and reported well, but the writing could be improved in places and should be checked throughout. I have a few comments.

Authors: Thank you for your positive feedback.

Reviewer #1: Abstract:

The conclusion in the abstract does not clearly represent the results of the meta-analysis, as it does not summarise the conclusion in the main paper. A summary could include something along the lines of “around a quarter of Ethiopians are colonised with ESBL-PE, and around one in 25 is colonised with CPE.” It should also make clear that it is based on a meta-analysis of studies judged to be of moderate-to-low risk of bias, and that the ESBL-PE study results were highly varied (high heterogeneity).

Authors: Thank you for your input. We have accepted your comments. We have revised the conclusion in both the abstract and the main document to ensure clarity. (Please refer to the revised manuscript's conclusion section in the abstract).

Reviewer #1: Introduction:

Line 82 – Should this be ESBL-PE rather than ESBLs?

Line 90 – It might be good to add that this is “in various specimens” here and add “using faecal specimens or swabs” to line 102.

Authors: Thank you for your positive feedback. We have rewritten these sentence as you modified (Please refer to the revised manuscript line # 85, 98-99 &105).

Reviewer #1: Lines 100 to 116 – It would be good to explain, somewhere, why this has changed from the protocol, which specifies hospitalised patients in Africa.

Authors: Thank you for reflecting on your concern. A systematic review was conducted on hospitalized patients worldwide, with results stratified by continent. Consequently, we have revised the protocol to focus on all patients in Ethiopia, removing restrictions on the study setting.

Reviewer #1: Methods:

Line 116 – see SF1 for a completed checklist.

Authors: Thank you for bringing this issue to our intention. We have checked and updated it to the 2020 PRISMA completed checklist (Please refer to the supplementary file 1 of revised manuscript).

Reviewer #1: Line 118 - Are these examples (electronic databases such as) or the full list of databases? All the sources searched should be clearly reported here or in the supplement.

Authors: Thank you for your positive feedback. We listed these electronic databases as examples because we attempted to search other electronic databases. However, now we are considering PubMed, Hinari, and Google Scholar as the complete list of databases since all 527 records were identified from these three sources (Please refer to the revised manuscript line # 112-113 & Fig 1).

Reviewer #1: Line 136 – the “No” doesn’t make sense here – should it be “Non-laboratory-based studies that reported…”?

Authors: Thank you for your input. We corrected a typing error, changing "No" to "All" because; we used laboratory-based studies. Please refer to the revised manuscript line # 131).

Reviewer #1: Line 156 – Was each record screened twice – once by each reviewer? How were differing decisions resolved? Or was each record screened by one reviewer?

Authors: Thank you for your positive feedback. The data were screened independently twice by two reviewers (MT and MA). Discrepancies were resolved through discussion, and unresolved differences were resolved by consulting a third reviewer (GG) (Please refer to the revised manuscript line # 148-150).

Reviewer #1: Line 170 – This sentence doesn’t make sense should it be …tools had 9 and 11 questions…?

Authors: Thank you for your input; we appreciate your feedback. We have accepted your comment and we tried to amend it. (Please refer to the revised manuscript line # 164-165).

Reviewer #1: Lines 171 to 172 – This sentence is not needed as this is explained in the next sentence.

Authors: Thank you for raising this interesting point. We have accepted your comment and removed this sentence (Please refer to the revised manuscript line # 165-166).

Reviewer #1: Lines 186 to 188 – It would be good to clearly report the changes to the protocol. The plan was for subgroups by region, strain and year of publication, so this adds setting, age and methods of outcome confirmation and removes strain and year of publication.

Authors: Thank you for reflecting on your concern. We included a subgroup analysis by year of publication; however, subgroup analysis by strain is less important, as it may be cumbersome and the pooled prevalence for each strain has already been performed (Please refer to the revised manuscript table 7 and supplementary file 3).

Reviewer #1: Results:

Lines 201 to 205 – There seems to be an error in the numbers reported here as 135 minus 96, 24 and 3 is 12 not 15. In the flow chart the 96 is 93.

Authors: Thank you for your input; we appreciate your feedback. We corrected a typing error; changing "96" to "93"(Please refer to the revised manuscript line # 189).

Reviewer #1: Discussion:

Line 361 – This sentence needs to be rewritten: CDT (31.2%) was comparable, and the lowest pooled estimate was…

Authors: Thank you for your input. We have re-written and modified the whole paragraph (Please refer to the revised manuscript line # 352-357).

Reviewer #1: Lines 378 to 383 - The conclusion in the paper is appropriate, but it could be written more clearly. It should also make clear that it is based on studies judged to be of moderate-to-low risk of bias, and that the ESBL-PE estimate has high heterogeneity.

Authors: Thank you for bringing this issue to our intention. We have revised the conclusion in both the abstract and the main document to ensure clarity (Please refer to the revised manuscript's conclusion section in the abstract and the main document).

Reviewer #1: Figure 1. This indicates that all the 527 records were identified from PubMed, Google Scholar and Hinari – were there none from the other databases or were they not searched?

Authors: Thank you for your positive feedback. We attempted to search other databases but did not incorporate them into our text and figures because our focus was on PubMed, Google Scholar, and Hinari, as these three major databases were sufficient. Therefore, all 527 records were identified from these three databases.

Reviewer #1: General:

Just a question, as I’m not a statistics expert, but is the probability reported the p-value for the heterogeneity or for the effect? For example, in Figure 3, the I-squared is zero, so no heterogeneity, the p-value is not significant (which is true for no significant heterogeneity), but the diamond doesn’t cross zero (the line of no effect) so should the p-value for the effect be significant?

Authors: Thank you for your comments and questions. I² is zero, indicating no heterogeneity among the studies. The p-value for heterogeneity is not significant, supporting no significant heterogeneity. The diamond does not cross zero, suggesting a statistically significant effect (the confidence interval does not include the line of no effect). If the diamond does not cross zero, it means the confidence interval does not include the null effect (typically "no effect" is represented by zero for mean differences or one for ratios). This implies that the result is statistically significant. Generally, the lack of heterogeneity (I² = 0 and non-significant heterogeneity p-value) strengthens the conclusion, as it suggests consistency across studies and no distortion of the pooled effect.

Reviewer #1: Were there any studies that did not meet the quality criterion? It seems that all fifteen studies were at moderate or low risk of bias, so I assume that no studies were actually excluded due to a high risk of bias? If this is the case, it might be good to state it clearly at the beginning of the results section, with a summary of the risk of bias results. I hope that my comments are helpful.

Authors: Thank you for raising this interesting point. We incorporated the risk of bias results at the beginning of the results section. All fifteen studies were at moderate or low risk of bias; therefore, no studies were excluded due to a high risk of bias (Please refer to the revised manuscript line # 196-199).

Comments raised by Reviewer 2

Reviewer #2: 1: A few places, the wording is unclear:

Line 136-139 – No laboratory-based studies have reported the presence of extended-spectrum beta-lactamase-producing and/or carbapenemase-producing Enterobacterales in human fecal specimens or rectal/perirectal swabs without restriction on publication year, and studies have been conducted in Ethiopia.

Authors: Thank you for your positive feedback. We corrected a typing error, changing "No" to "All" because; we used laboratory-based studies (Please refer to the revised manuscript line # 131).

Reviewer #2: Line 170 – The quality assessment tools were 9 and 11 for prevalence and cohort studies, respectively.

Authors: Thank you for your input. We have accepted your comment and we tried to amend it. (Please refer to the revised manuscript line # 164-165).

Reviewer #2: Line 378-383– Conclusion and recommendations section

Authors: Thank you for bringing this issue to our intention. We have revised the conclusion in both the abstract and the main document to ensure clarity (Please refer to the revised manuscript's conclusion section in the abstract and the main document).

Reviewer #2: Line 202 and flow chart: There is a mismatch between the flow chart and the text. When doing the calculations, the flowchart match perfectly, by subtracting 93 (43 + 36 + 14) from 135, with 42 studies left. In the text, however, 93 has been replaced by 96. 96 should probably be 93, then everything fits perfectly.

Authors: Thank you for your input; we appreciate your feedback. We corrected a typing error; changing "96" to "93"(Please refer to the revised manuscript line # 189).

Comments raised by Reviewer 3

Reviewer #3: The systematic review appeared to be fairly comprehensive with search strategy, eligibility checks, main outcomes checked of the colonization rate of ESBL-PE and CPE from human fecal/rectal swab specimens in Ethiopia, data extraction and management outlines according to PRISMA and quality of study assessment using standard critical appraisal tools prepared by the Joanna Briggs Institute (JBI) for prevalence and cohort studies as is the case in this application. The data analysis endpoint was mainly colonization rate of ESBL-PEs and CPEs determined separately via a random effects model which is certainly appropriate in this context. Heterogeneity was assessed via the I square statistic. Sensitivity analysis was conducted to determine the influence of single studies on the pooled estimates. Publication bias and small study effects were checked by using a funnel plot test and Egger’s test. It appears that all the elements of this systematic review and meta-analysis were in place. The manuscript was well organized.

Authors: Thank you for your positive feedback.

Reviewer #3: When evaluating the publication bias the authors note that the ESBL-PE analysis revealed an asymmetric funnel plot, which indicates the presence of publication bias (Fig. 4). Similarly, Egger’s test revealed significant publication bias (P=0.004), as indicated in Table 3. Also in Fig 5. the Funnel plot showed publication biases related to the colonization of CPE. The investigators should explain more completely, if possible, the sources of the asymmetry in both plots.

Authors: Thank you for bringing this issue to our intention. We have performed a trim-and-fill analysis to address the observed publication bias (Please refer to the revised manuscript line # 264-269, table 4 and 5).

Reviewer #3: Also the interpretation of sensitivity is not detailed sufficiently for the reader in Table 4. In addition, in Table 5, the possible sources of heterogeneity, especially for those with an I squared above 90 and statistically significant (p<0.05) should be explained, if possible, for the reader.

Authors: Thank you for reflecting on your concern. We have sought to interpret the sensitivity analysis with sufficient detail (Please refer to the introduction portion of revised manuscript line # 275-278).

Reviewer #3: The investigators use the statistic, ES, on the Forest plots throughout the manuscript and in supplemental file 3. Explain that it is the colonization rate. If otherwise, please define it. Also the manuscript should be edited to be sure the reader knows what the p-values are measuring, much like the authors did for explaining the p-values associated with heterogeneity and publication bias.

Authors: Authors: We appreciate your feedback. We have added colonization rate (Please refer to the revised manuscript)

Authors: Finally, we would like to say thank you for reviewing our work and making insightful suggestions and comments that helped to strengthen our manuscript. We have revised the manuscript as necessary.

---

## [Decision Letter · Decision Letter 1]

2 Feb 2025

PONE-D-24-33532R1Colonization with extended-spectrum β-lactamase and carbapenemase-producing Enterobacterales in Ethiopia: a systematic review and meta-analysisPLOS ONE

Dear Dr. Tigabie,

Thank you for submitting your manuscript to PLOS ONE. After careful consideration, we feel that it has merit but does not fully meet PLOS ONE’s publication criteria as it currently stands. Therefore, we invite you to submit a revised version of the manuscript that addresses the points raised during the review process. Thank you very much for what you have done. The article is almost ready to be published. I ask you to respond to the new comments of Reviewer 1.

We look forward to receiving your revised manuscript.

Kind regards,

Lorenzo Righi

Academic Editor

PLOS ONE

Journal Requirements:

Reviewers' comments:

Reviewer's Responses to Questions

**Comments to the Author**

1. If the authors have adequately addressed your comments raised in a previous round of review and you feel that this manuscript is now acceptable for publication, you may indicate that here to bypass the “Comments to the Author” section, enter your conflict of interest statement in the “Confidential to Editor” section, and submit your "Accept" recommendation.

Reviewer #1: (No Response)

Reviewer #3: All comments have been addressed

2. Is the manuscript technically sound, and do the data support the conclusions?

Reviewer #1: Partly

Reviewer #3: (No Response)

3. Has the statistical analysis been performed appropriately and rigorously? 

Reviewer #1: I Don't Know

Reviewer #3: (No Response)

4. Have the authors made all data underlying the findings in their manuscript fully available?

Reviewer #1: Yes

Reviewer #3: (No Response)

5. Is the manuscript presented in an intelligible fashion and written in standard English?

Reviewer #1: No

Reviewer #3: (No Response)

6. Review Comments to the Author

Reviewer #1: Thank you for addressing my comments. There are two queries that could be clarified further (points 1 and 2 below), and five minor points (3 to 7 below).

1. The p-values that are reported in brackets after the effect estimates appear to be probabilities for the I-squared heterogeneity measure. For example, 4.4% (95% CI: 0.9–7.9%, P = 0.636) – the effect estimate probability is 0.01, while the I-squared probability is 0.64.

ChatGPT provided this example to explain how the probability for I-squared should be distinguished from the probability for the effect estimate:

The pooled odds ratio (OR) for the intervention’s effect on mortality was 0.78 (95% CI: 0.65–0.93, p = 0.004), indicating a significant reduction in risk. However, substantial heterogeneity was observed (I² = 67%, p < 0.01), suggesting variation among studies.

If you use the p for heterogeneity in brackets after the effect estimate, at a minimum, it requires I² before it, otherwise it appears to be the p for the effect estimate.

2. As required by item 24c in the PRISMA checklist, any changes to those methods that were prespecified in the protocol, should be reported in the paper or a supplementary file.

3. Line #131 has been revised, but it is not a sentence. It should start with “We included…” or should contain a statement that these studies were included.

4. In Table 1, what does the “no of cases” refer to? How does this differ from the Total ESBL-PE? How can the number of cases be fewer than the total ESBL-PE (see the first line, Temsegen)?

5. Lines 303 and 358 – according to the figure, the 55.22 should be 55.25.

6. Line 345 - when comparing EBSL-PE and CPE in brackets it should be (29.9% and 4.6%) rather than vs to match the text where the comparison is between hospital versus community.

7. Line 351 - I think this should be ESBL-PE rather than just ESBL.

Otherwise, everything has been addressed, thank you.

Reviewer #3: (No Response)

7. PLOS authors have the option to publish the peer review history of their article (what does this mean? ). If published, this will include your full peer review and any attached files.

**Do you want your identity to be public for this peer review?** For information about this choice, including consent withdrawal, please see our Privacy Policy .

Reviewer #1: **Yes: ** Claire Khouja

Reviewer #3: No

---

## [Author Response · Author response to Decision Letter 2]

6 Feb 2025

Response to Academic editor and Reviewers

Authors: We appreciate for spending your precious time and forwarding your valuable comments, which have significantly improved our manuscript. We are also grateful for this positive feedback. Please see below, bold, for a point-by-point response to the academic editor and reviewers. We've copied your comments and responses below to make things easier for you. All line numbers refer to the revised manuscript file.

Academic editor:

Journal Requirements: Please review your reference list to ensure that it is complete and correct. If you have cited papers that have been retracted, please include the rationale for doing so in the manuscript text, or remove these references and replace them with relevant current references. Any changes to the reference list should be mentioned in the rebuttal letter that accompanies your revised manuscript. If you need to cite a retracted article, indicate the article’s retracted status in the References list and also include a citation and full reference for the retraction notice.

Authors: Thank you for your positive feedback. We have amended the reference list according to PLOS ONE’s criteria, and we ensure that it is complete and correct.

Reviewer #1: Thank you for addressing my comments. There are two queries that could be clarified further (points 1 and 2 below), and five minor points (3 to 7 below).

Authors: Thank you for your input. We have accepted your comments and revised the entire document accordingly. We appreciate the opportunity to learn from your feedback.

Reviewer #1: Thank you for addressing my comments. There are two queries that could be clarified further (points 1 and 2 below), and five minor points (3 to 7 below).

1. The p-values that are reported in brackets after the effect estimates appear to be probabilities for the I-squared heterogeneity measure. For example, 4.4% (95% CI: 0.9–7.9%, P = 0.636) – the effect estimate probability is 0.01, while the I-squared probability is 0.64.

ChatGPT provided this example to explain how the probability for I-squared should be distinguished from the probability for the effect estimate:

The pooled odds ratio (OR) for the intervention’s effect on mortality was 0.78 (95% CI: 0.65–0.93, p = 0.004), indicating a significant reduction in risk. However, substantial heterogeneity was observed (I² = 67%, p < 0.01), suggesting variation among studies.

If you use the p for heterogeneity in brackets after the effect estimate, at a minimum, it requires I² before it, otherwise it appears to be the p for the effect estimate.

Authors: The p-values reported in brackets after the effect estimates represent the probabilities for the I-squared heterogeneity measure. To clarify, we have added "I²" before all p-values. Please refer to the revised manuscript's abstract and results sections.

Reviewer #1: 2. As required by item 24c in the PRISMA checklist, any changes to those methods that were prespecified in the protocol, should be reported in the paper or a supplementary file.

Authors: Thank you for your input. We have accepted your comments and revised the Methods section to report the deviations from the original protocol. Please refer to the revised manuscript (lines 111–125).

Reviewer #1: 3. Line #131 has been revised, but it is not a sentence. It should start with “We included…” or should contain a statement that these studies were included.

Authors: Thank you for your positive feedback. We have rewritten these sentence as you modified (Please refer to the revised manuscript line # 146).

Reviewer #1: 4. In Table 1, what does the “no of cases” refer to? How does this differ from the Total ESBL-PE? How can the number of cases be fewer than the total ESBL-PE (see the first line, Temsegen)?

Authors: Thank you for reflecting on your concern. We attempt to clarify this issue: the number of cases refers to the number of participants. The number of cases may be lower than the number of ESBL-PE and CPE because some study participants may be colonized by two or more ESBL-PE and CPE.

Reviewer #1: 5. Lines 303 and 358 – according to the figure, the 55.22 should be 55.25.

Authors: Thank you for bringing this issue to our intention. We have checked and corrected it to the 55.22 to 55.25 (Please refer to the revised manuscript line # 52, 319 & 374).

Reviewer #1: 6. Line 345 - when comparing EBSL-PE and CPE in brackets it should be (29.9% and 4.6%) rather than vs to match the text where the comparison is between hospital versus community.

Authors: Thank you for your positive feedback. We corrected this language error (Please refer to the revised manuscript line # 361).

Reviewer #1: 7. Line 351 - I think this should be ESBL-PE rather than just ESBL.

Authors: Thank you for your input. We corrected a typing error, changing "ESBL" to "ESBL-PE" (Please refer to the revised manuscript line # 367).

Authors: Finally, we would like to say thank you for reviewing our work and making insightful suggestions and comments that helped to strengthen our manuscript. We have revised the manuscript as necessary.

---

## [Editor Report · Decision Letter 2]

10 Feb 2025

Colonization with extended-spectrum β-lactamase and carbapenemase-producing Enterobacterales in Ethiopia: a systematic review and meta-analysis

PONE-D-24-33532R2

Dear Dr. Tigabie,

We’re pleased to inform you that your manuscript has been judged scientifically suitable for publication and will be formally accepted for publication once it meets all outstanding technical requirements.

Kind regards,

Lorenzo Righi

Academic Editor

PLOS ONE
---

## [Editor Report · Acceptance letter]

PONE-D-24-33532R2

PLOS ONE

Dear Dr. Tigabie,

I'm pleased to inform you that your manuscript has been deemed suitable for publication in PLOS ONE. Congratulations! Your manuscript is now being handed over to our production team.

Kind regards,

on behalf of

Dr. Lorenzo Righi

Academic Editor

PLOS ONE
